# Potential of Carbohydrate Antigen 19-9 and Serum Apolipoprotein A2-Isoforms in the Diagnosis of Stage 0 and IA Pancreatic Cancer

**DOI:** 10.3390/diagnostics14171920

**Published:** 2024-08-30

**Authors:** Keiji Hanada, Akihiro Shimizu, Ken Tsushima, Michimoto Kobayashi

**Affiliations:** 1Department of Gastroenterology, Onomichi General Hospital, Onomichi 722-8508, Japan; a.shimizu313@gmail.com (A.S.); k.tsushima@onomichi-gh.jp (K.T.); 2Toray Industries, Inc., Tokyo 103-8066, Japan; michimoto.kobayashi.n6@mail.toray

**Keywords:** apoA2-ATQ/AT, early diagnosis, biomarker, pancreatic cancer, CA19-9

## Abstract

Apolipoprotein A2-ATQ/AT (apoA2-ATQ/AT) is a new biomarker for diagnosing pancreatic cancer (PC). In this study, the value of blood carbohydrate antigen 19-9 (CA19-9) and apoA2-ATQ/AT levels in diagnosing stage 0 and IA PC was evaluated. During 2014–2021, 12 patients with stage 0 PC and 12 patients with IA PC (average age: 73.8 years) underwent resection at JA Onomichi General Hospital. In addition, the data of 200 healthy controls were collected from a community-based cohort study. Levels of two apoA2-isoforms were measured using enzyme-linked immunosorbent assay (ELISA) with specific antibodies to calculate the apoA2-i Index as a surrogate value for apoA2-ATQ/AT. The cutoff value for the apoA2-i Index was determined to be 62.9 μg/mL. CA19-9 levels were also measured through ELISA. Among all 24 patients with PC, the positivity rates for apoA2-i and CA19-9 were 33.3% and 25.0%, respectively. The positivity rates for apoA2-i and CA19-9 were 16.7% and 8.3% in patients with stage 0 PC and 50.0% and 41.7% in those with stage IA, respectively. For CA19-9-negative patients, the apoA2-i positivity rate was 9.1% in stage 0 and 42.9% in stage IA. The combined positivity rate for both markers was 16.7% in stage 0 and 66.7% in stage IA. Imaging findings in apoA2-i- and CA19-9-positive patients included pancreatic duct dilatation (87.5%/100%), duct stenosis (75.0%/50%), and atrophy (87.5%/66.7%). The imaging findings of this study suggest that apoA2-i may enhance the sensitivity for detecting CA19-9-negative stage 0 and IA PC, and complementary measurements with CA19-9 may be valuable for diagnosing early-stage PC. Therefore, minute PC with pancreatic duct dilation, duct stenosis, and atrophy may exhibit a high positivity rate, aiding differential diagnosis.

## 1. Introduction

Pancreatic cancer (PC) is one of the leading causes of cancer-related deaths worldwide [1]. Early detection of PC is crucial for improving patient prognosis; however, identifying PC in its early stages is challenging. A recent analysis by the Japan Pancreas Society of the Japan Pancreatic Cancer Registry revealed that patients with a PC of <10 mm in size had a 5-year survival rate of 80.4%, whereas patients classified as having the Union for International Cancer Control (UICC) stage 0 pancreatic ductal adenocarcinoma, comprising high-grade pancreatic intraepithelial neoplasia (HG-PanIN)/pancreatic carcinoma in situ, reported a survival rate of 85.8% [2]. Nonetheless, patients with UICC stage 0 and IA pancreatic ductal adenocarcinoma represented only 1.7% and 4.1%, respectively [2]. Given the deep location of the pancreas within the abdomen, a combination of diagnostic imaging modalities, including ultrasonography (US), computed tomography (CT), magnetic resonance imaging (MRI), endoscopic ultrasonography (EUS), and endoscopic retrograde cholangiopancreatography (ERCP), is necessary for early-stage PC diagnosis. Developing efficient methods, including potential blood biomarkers, for screening the general population is crucial for identifying early-stage PC [3].

Carbohydrate antigen 19-9 (CA19-9) is a traditional biomarker used for detecting PC and monitoring therapy response in patients with PC [4]. However, CA19-9 levels can also be elevated in advanced biliary and gastrointestinal cancers, as well as certain benign conditions, such as hepatic cysts [5]. Notably, individuals genetically expressing nonsialylated Lewis blood group antigens do not express CA19-9 at all [6]. Furthermore, studies have shown that the sensitivity of CA19-9 for early-stage PC, including stages 0 and I, is limited [7,8].

ApoA2 is a key component of high-density lipoproteins and plays an important role in directing lipid metabolism for these lipoproteins. Human apoA2 comprises 77 amino acids and mainly circulates as a dimer in the bloodstream [9]. ApoA2 protein is categorized into three types based on the C-terminal amino acid sequence of the monomer: ATQ, AT, and A types, resulting in five isoforms of the apoA2 homodimer: ATQ/ATQ, ATQ/AT, AT/AT, AT/A, and A/A [10]. Recent research has indicated that alterations in apoA2-ATQ/AT concentrations due to abnormal amino acid processing at the C-terminal end of the apoA2 homodimer can help in distinguishing patients with early-stage PC from healthy individuals [10,11,12,13]. In addition, the combined mean values of apoA2-AT and apoA2-TQ concentrations termed the apoA2-i Index can serve as a surrogate marker for apoA2-AT/TQ [13,14]. These findings suggest that apoA2-ATQ/AT holds promise as a valuable biomarker for detecting minute PC. However, studies assessing blood apoA2-ATQ/AT levels as a screening biomarker for early-stage PC, specifically stages 0 and I, while considering imaging findings from CT, MRI, and EUS, are currently lacking.

## 2. Materials and Methods

### 2.1. Patients and Samples

From 2014 to 2021, 24 cases of stage 0 and IA PC (classified according to the 8th edition of the UICC) were resected at JA Onomichi General Hospital. The average age of these 24 patients was 73.8 years, with a male-to-female ratio of 7:17. Of these patients, 12 had stage 0 PC, and 12 had stage IA PC. This study also included 200 healthy controls (100 men and 100 women, average age: 63.9 years) from the Tohoku Medical Megabank Project (research number: 2021-0045). The data of these 200 healthy controls were collected during a community-based cohort study (Tohoku Medical Megabank CommCohort Study) [15] from 2013 to 2016. The inclusion criteria were individuals aged between 49 and 80 years, whereas the exclusion criteria were individuals with diabetes, pancreatitis, and a history of malignant neoplasms. Serum samples were collected under the standard operating procedure of each clinical institution. All participants agreed to sample collection and provided written informed consent.

### 2.2. Methods

#### 2.2.1. Laboratory Assays

The apoA2-i ELISA kit (Toray Industries, Inc., Tokyo, Japan) was developed to meet the requirements of the Japanese medical device quality management system. (Figure 1) [11]. In total, 224 subjects were enrolled in this study, and their serum apoA2-AT and apoA2-TQ levels were measured in a single assay under the instruction manual of the kit.

Each serum sample was diluted 10,000-fold with sample diluent and added to the wells of the ELISA plate coated with anti-apoA2 antibody to measure apoA2-AT and apoA2-TQ separately. After incubation for 30 min, the antigen in the sample was captured by the antibodies on the surface of the wells. After washing and removing the unreacted specimen, horseradish peroxidase-labeled antibodies for apoA2-AT or apoA2-TQ were added to create an antibody–antigen–enzyme complex. After washing off any unreacted antibodies, 100 µL of 3,3′,5,5′-tetramethylbenzidine substrate was added to each well and incubated at room temperature for 30 min. The reaction was stopped with a sulfuric acid solution, and the absorbance of each well was measured at 450 nm (with a reference wavelength of 650 nm). The concentrations of the two apoA2 isoforms in the sample were calculated using a four-parameter logistic curve generated from the absorbance of the standard solutions measured simultaneously.

The apoA2-i Index was calculated using the following formula:$$\mathrm{ApoA2-i}\;Index=\sqrt{\left(apoA2-TQ\right)\times (apoA2-AT)}$$

If the measurements fell below the lower limit of quantification (apoA2-AT concentration: <3.25 μg/mL and apoA2-TQ concentration: <5.75 μg/mL), they were considered to have an apoA2-i Index = of zero (=positive). The cutoff value for the apoA2-i Index (62.9 μg/mL) was determined based on the distribution of values in healthy samples, with a negative rate of 95%. CA19-9 levels were measured using an established Lumipulse Presto CA19-9 Kit (Fujirebio, Inc., Tokyo, Japan). In this study, samples of all patients had CA19-9 values below the detection limit value of 2 U/mL; hence, this lower threshold value was assigned to all samples.

#### 2.2.2. Imaging Diagnosis of Stage 0 and IA PC

For the imaging diagnosis of PC in all patients, enhanced CT, MRI, and EUS were conducted. For preoperative pathological diagnosis, EUS-guided fine-needle aspiration or biopsy was performed for small mass lesions in the pancreas. ERCP with pancreatic fluid cytology was conducted multiple times in patients with specific pancreatic duct findings and no mass lesions [3].

After all imaging diagnoses, the presence or absence of tumor lesions was determined using EUS and enhanced CT. A mass-like lesion was assessed for its diameter and contrast enhancement pattern. The dilatation and stenosis of the main pancreatic duct (MPD) were assessed using MRI and magnetic resonance cholangiopancreatography (MRCP). Pancreatic duct dilatation was defined as a maximum diameter of the MPD of >3 mm. Atrophic changes in the pancreatic parenchyma were categorized as “normal”, “wide”, and “local” based on enhanced CT scans. Wide atrophy was defined as atrophy throughout the entire pancreas without focal lesions. Local parenchymal atrophy was defined as parenchymal atrophy in the distal part of the pancreatic focal lesion or disproportional atrophy in the absence of a pancreatic focal lesion [16].

#### 2.2.3. Pathological Diagnosis of Stage 0 and IA PC

Among the 12 patients with stage 0 PC, 9 had HG-PanIN and 3 had noninvasive intraductal papillary mucinous carcinoma. Among the 12 patients with stage IA PC, 8 had well-differentiated adenocarcinoma, 2 had moderately differentiated adenocarcinoma, and 2 had intraductal papillary mucinous neoplasm (IPMN)-derived invasive adenocarcinoma.

### 2.3. Data Analysis

All data analyses were performed using Microsoft^®^ Excel^®^ for Microsoft 365 MSO version 2108 (Microsoft Corporation, Redmond, WA, USA).

## 3. Results

### 3.1. Positivity Rates for the apoA2-i Index and CA19-9 by PC Stage

In total, 224 patients with PC and healthy individuals were included in this study and subjected to apoA2-AT and apoA2-TQ measurements using the apoA2-i ELISA kit. The analysis of the positivity rate in stage 0 and IA PC was performed, as shown in the flow chart in Figure 2. The positivity rates for the apoA2-i Index and CA19-9 in patients with stage 0 and IA PC were 33.3% and 25.0%, respectively. Notably, the positivity rate of the apoA2-i Index surpassed that of CA19-9 in patients with stage 0 and IA PC. In addition, the positivity rate for the apoA2-i Index in CA19-9-negative patients was 9.1% for stage 0 PC and 42.9% for stage IA PC (Table 1 and Figure 3). The two patients with stage 0 PC and positive for the apoA2-i Index both had noninvasive cancers derived from IPMN. In contrast, the ten patients with stage 0 PC corresponding to HG-PanIN were negative for the apoA2-i Index.

### 3.2. Positivity Rates for the apoA2-i Index and CA19-9 by Stage 0 PC

The positivity rates for the apoA2-i Index and CA19-9 in PC patients were both 0%, respectively, in those with intraductal tumor spread less than 5 mm and 25.0% and 12.5%, respectively, in PC cases with intraductal tumor spread ranging from 5 to 10 mm. The positivity rate of the apoA2-i Index surpassed that of CA19-9 in PC cases with intraductal tumor spread ranging from 5 to 10 mm (Table 1 and Figure 4).

### 3.3. Positivity Rates for the apoA2-i Index and CA19-9 by Stage IA PC

The positivity rates for the apoA2-i Index and CA19-9 in PC patients were both 25%, respectively, with tumor sizes < 10 mm and 62.5% and 50.0%, respectively, in those with tumor sizes ranging from 10 to 20 mm. The positivity rate of the apoA2-i Index surpassed that of CA19-9 in PC patients with tumor sizes ranging from 10 to 20 mm (Table 1 and Figure 5).

### 3.4. Positivity Rates with a Combination of the apoA2-i Index and CA19-9

The positivity rate with a combination of the apoA2-i Index and CA19-9 was 16.7% in patients with stage 0 PC and 66.7% in those with stage IA PC. Combining measurements of the apoA2-i Index and CA19-9 may aid in the diagnosis of early-stage PC (Table 1).

### 3.5. Association between Imaging Findings and the apoA2-i Index and CA19-9

In this study, eight patients with PC tested positive for the apoA2-i Index, and six tested positive for CA19-9. Indirect imaging findings, such as MPD dilatation, MPD stenosis, and local pancreatic atrophy, were identified using enhanced CT, MRI, and EUS. Among the eight patients with PC positive for the apoA2-i Index, MPD dilatation was detected in seven (87.5%), MPD stenosis in six (75%), and local pancreatic atrophy in seven (87.5%) (Figure 6). Of the six patients with PC positive for CA19-9, MPD dilatation was detected in six (100%), MPD stenosis in three (50%), and local pancreatic atrophy in four (66.7%). Overall, patients with apoA2-i Index-positive PC tended to exhibit a higher incidence of MPD stenosis and local pancreatic atrophy (Table 1).

### 3.6. Association between Pathological Findings and the apoA2-i Index and CA19-9

In stage 0 PC, the positivity rates of the apoA2-i Index and CA19-9 were 66.7% and 33.3%, respectively, in cases of IPMN with high-grade dysplasia, while both were negative in cases of HG-PanIN. Moving to stage IA PC, the positivity rates of the apoA2-i Index and CA19-9 were 50% and 40%, respectively, in cases of invasive ductal PC. Notably, one case of IPMN-derived invasive carcinoma was positive for CA19-9, and another case of intraductal tubulopapillary neoplasm-derived invasive carcinoma was positive for the apoA2-i Index (Table 1).

## 4. Discussion

The prognosis for patients with PC has been historically poor due to the challenges of early detection of small tumors within the deep-seated pancreas during routine examinations. Early diagnosis is crucial for improving the prognosis of patients with PC [1,17]. In Japan, the clinical practice guidelines for PC suggest that for the diagnosis of early-stage PC, if a small mass lesion is identified by EUS, EUS-guided fine-needle aspiration (EUS-FNA) should be performed. If no mass is observed but imaging shows signs suspicious of minute PC, such as abnormalities in the pancreatic duct, ERCP and pancreatic juice cytology should be complementarily conducted [3,18].

Serum tumor marker measurement is a simple and minimally invasive method for diagnosis and is considered a primary diagnostic tool. Among these markers, the glycoprotein CA19-9 has long been recognized as an important marker for PC diagnosis [19]. However, it is limited as a screening tool for detecting early-stage PC in asymptomatic patients. The positivity rate of tumor markers in patients with stage 0 PC has been reported to be 2.7% for carcinoembryonic antigen, 11% for CA19-9, 4.8% for Duke pancreatic monoclonal antigen type 2, and 10% for s-pancreas antigen-1 [8]. These findings indicate that elevated levels of tumor markers are rarely detected in patients with early-stage PC in clinical practice. The development of novel tumor markers in combination with existing ones holds promise for improving early-stage PC diagnosis.

Kashiro et al. conducted a blind verification of their proprietary apoA2-i ELISA to discriminate stage I/II PC cases from healthy subjects using the PC plasma reference set from the National Cancer Institute Early Detection Research Network. The point estimate of the area under the curve for apoA2-ATQ/AT was found to be higher than that for CA19-9, and combining CA19-9 with apoA2-ATQ/AT led to a significantly higher area under the curve than that obtained using CA19-9 alone [11].

The mechanism underlying the over-cleavage or inhibition of cleavage of the C-terminus of apoA2-i in PC has been speculated as follows: Under normal pancreatic function, apoA2-i is cleaved into apoA2-ATQ/ATQ, apoA2-ATQ/AT, and apoA2-AT/AT. However, PC, predominantly pancreatic ductal carcinoma, originates from the pancreatic duct and the drainage duct of pancreatic juice, often with a normal diameter of approximately 2 mm. It is suggested that the accumulation of pancreatic juice in the pancreatic duct due to very small PC may lead to increased intrapancreatic pressure and carboxypeptidase leakage into the bloodstream, thereby enhancing the cleavage of the C-terminal side of apoA2-i. Decreased exocrine pancreatic function also reduces carboxypeptidase production, likely resulting in suppressed apoA2-i cleavage [20,21,22]. To date, no studies have assessed the utility of apoA2-i in combination with CA19-9 in diagnosing stage 0 and IA PC with a favorable long-term prognosis.

In the present study, the positivity rates for apoA2-i and CA19-9 in patients with stage 0 and IA PC were 33.3% and 25.0%, respectively. Notably, the positivity rate for apoA2-i surpassed that for CA19-9 in patients with stage 0 and IA PC with tumor diameters < 20 mm. The apoA2-i positivity rate for patients with CA19-9-negative stage 0 and IA PC was 22.2%, which increased to 41.7% when combined with CA19-9. These findings indicate that concurrent measurements of apoA2-i and CA19-9 may enhance the diagnosis of very small PC.

Interestingly, patients with apoA2-i-positive PC showed a high incidence of pancreatic duct dilation, pancreatic duct stenosis, and local pancreatic atrophy. Recent multicenter studies conducted in Japanese high-volume centers revealed that the sensitivity of tumor detection using US, CT, and MRI was low for patients with stage 0 and IA PC. In contrast, EUS could detect tumorous lesions in approximately 25% of patients with stage 0 PC and 80% of those with stage IA PC [7,8]. Indirect imaging findings, including MPD dilatation, were frequently detected on US, CT, MRI, including MRCP, and EUS in 70–80% of patients with stage 0 and IA PC. MPD stenosis was frequently detected in 80% of patients subjected to MRCP and EUS [7,8,23,24,25]. Recent studies reported that CT scans can detect local pancreatic atrophy or focal fatty changes in parenchyma in 30–64% of patients with stage 0 PC [7,8,26,27]. The temporal progression of focal parenchymal atrophy is the earliest detectable sign indicating early-stage PC [28]. These findings suggest that MPD dilatation, MPD stenosis, and local pancreatic atrophy are crucial initial indicators for the early diagnosis of PC.

Based on the results of this study, 25% of patients with stage 0 PC, which shows progression of 5 mm or more within the pancreatic duct, were positive for the apoA2-i Index, demonstrating a better sensitivity compared to CA19-9. It has been reported that patients with stage 0 PC, in which the MPD is stenotic, as seen on MRI and other imaging diagnostics, often reflect progression within the MPD when compared with resected specimens [23]. Therefore, the apoA2-i Index may have the potential to screen for patients with early-stage PC, where there is no detectable tumorous lesion, but the pancreatic duct is stenotic. Screening for early-stage PC by measuring apoA2-i in patients with these indirect pancreatic imaging abnormalities may be feasible based on the hypothesis shown in Figure 7. However, prospective studies involving a larger number of cases are necessary to further investigate this potential screening approach.

Recently, in Japan, collaborative efforts between hospitals and clinics have been initiated to promote early diagnosis of PC [29]. Since 2007, the Onomichi Medical Association, along with our institution, has been involved in the Onomichi Project, which focuses on cooperation between core and affiliated facilities. For patients with abdominal symptoms, risk factors, or abnormal blood test results, US is mainly performed at the affiliated facilities. If indirect findings, such as pancreatic cysts or MPD dilatation, are detected, patients are referred to the core facility for further evaluation using MRCP or EUS. As a result of the Onomichi Project, a total of 18,507 suspicious cases of PC were referred to the core facility between 2007 and 2020, and after detailed examinations, 610 cases were pathologically diagnosed with the disease. Among them, 32 cases were diagnosed at stages 0 and I, indicating an increase in early diagnosis, higher surgical resection rates, and better 5-year survival rates [30]. Similar initiatives have been launched in other regions of Japan, such as Osaka, Kishiwada [31], Matsue, and Wakayama, tailored to the specific medical needs of each area. Some regions have reported outcomes comparable to those in the Onomichi region. In the future, the usefulness of apoA2-i in early PC screening is expected to be evaluated in these domestic projects to enhance early-stage diagnosis of this disease in Japan.

Recently, Felix et al. reported that the apoA2-i Index holds promise for the early detection of malignancy risk in patients with IPMN [32]. Their study results revealed a positivity rate for the apoA2-i Index of 66.7% in patients with IPMN with high-grade dysplasia, suggesting its potential utility in the early diagnosis of malignant lesions arising from IPMN. However, all 10 patients with HG-PanIN had a negative apoA2-i Index, indicating potential limitations in the diagnosis of this condition. Further investigations involving a larger number of cases are required to elucidate the diagnostic potential of the apoA2-i Index for HG-PanIN.

Chronic inflammation might affect the serum levels of apoA2-i. Not examined in this study, the subpopulation of patients with chronic pancreatitis could not be distinguished from PC by blood testing for apoA2-i. Honda et al. reported that the AUC value to discriminate healthy controls and chronic pancreatitis by apoA2-i was 0.992 [13]. Furthermore, a recent study using two independent cohorts from Japan and the United States reported apoA2-ATQ/AT positivity rates of 40% and 50% in chronic pancreatitis [11]. The diagnosis of PC is expected to be based on a comprehensive consideration of the results from biomarker tests, including apoA2-i and diagnosis imaging tests (e.g., enhanced CT, MRI, and EUS).

## 5. Conclusions

In conclusion, apoA2-i may enhance the sensitivity of detecting CA19-9-negative stage 0 and IA PC, and complementary measurements with CA19-9 could be beneficial for diagnosing early-stage PC. In particular, cases of minute PC with pancreatic duct dilation, stenosis, and pancreatic atrophy may exhibit a high positivity rate, which could aid in differential diagnosis.

## Figures and Tables

**Figure 1 diagnostics-14-01920-f001:**
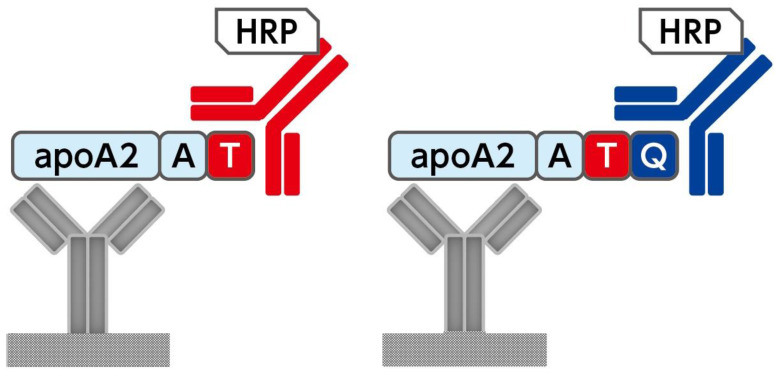
ApoA2-i enzyme-linked immunosorbent assay (ELISA) kit. In sandwich ELISAs, apoA2-AT and apoA2-TQ analytes are bound to microplate wells using anti-apoA2 non-terminal antibodies and detected with horseradish-peroxidase (HRP)-labeled anti-apoA2 C-terminal-specific antibodies for apoA2-AT or apoA2-TQ.

**Figure 2 diagnostics-14-01920-f002:**
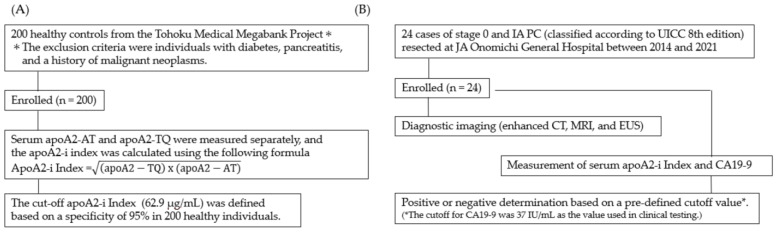
Flow chart of the experimental procedure. (**A**) Healthy individuals in the cutoff test (*n* = 200). Individuals with diabetes, pancreatitis, and a history of malignant neoplasm were excluded from this study. (**B**) Pancreatic cancers of stage 0 and IA (*n* = 24) for evaluating the clinical performance of apoA2-i Index and CA19-9. For determination by CA19-9, the cutoff value used in clinical testing (37 IU/mL) was used.

**Figure 3 diagnostics-14-01920-f003:**
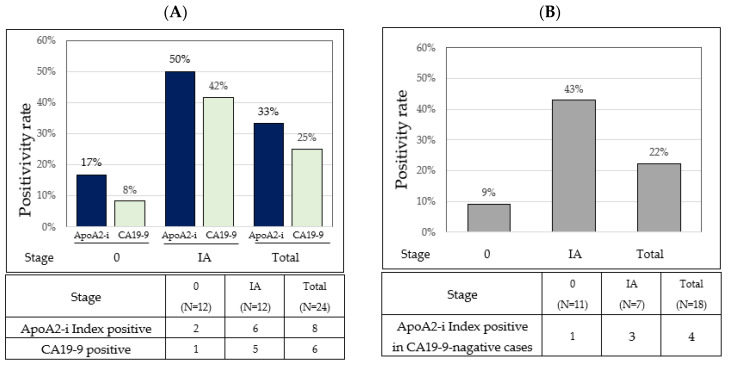
Positivity rates for the apoA2-i Index and CA19-9 by PC stage. (**A**) Positive rates (**upper**) and number of positives (**lower**) for apoA2-i Index and CA19-9 in each stage of PC are shown. (**B**) Positive rates (**upper**) and number of positives (**lower**) for apoA2-i Index in each stage of CA19-9-negative cases are shown.

**Figure 4 diagnostics-14-01920-f004:**
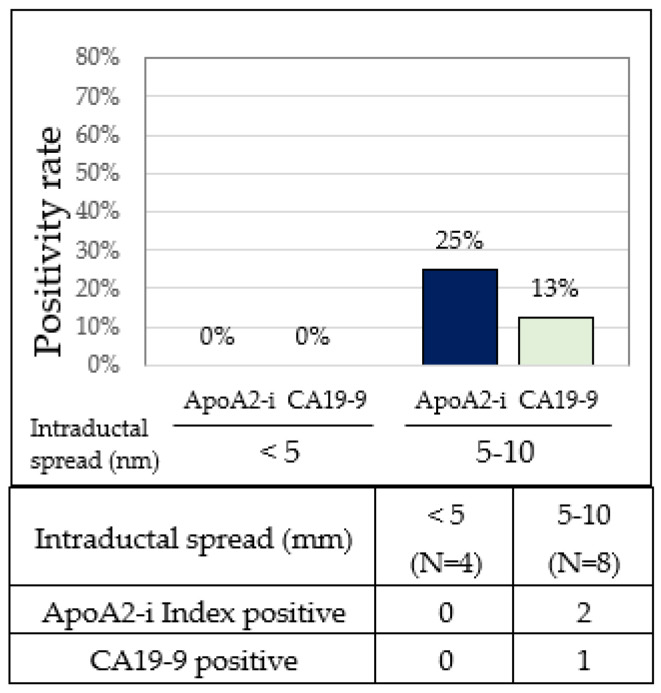
Positivity rates of apoA2-i Index and CA19-9 by stage 0 PC. Positivity rates (**upper**) and number of positives (**lower**) for apoA2-i Index and CA19-9 in pancreatic cancers with different degrees of intraductal tumor spread.

**Figure 5 diagnostics-14-01920-f005:**
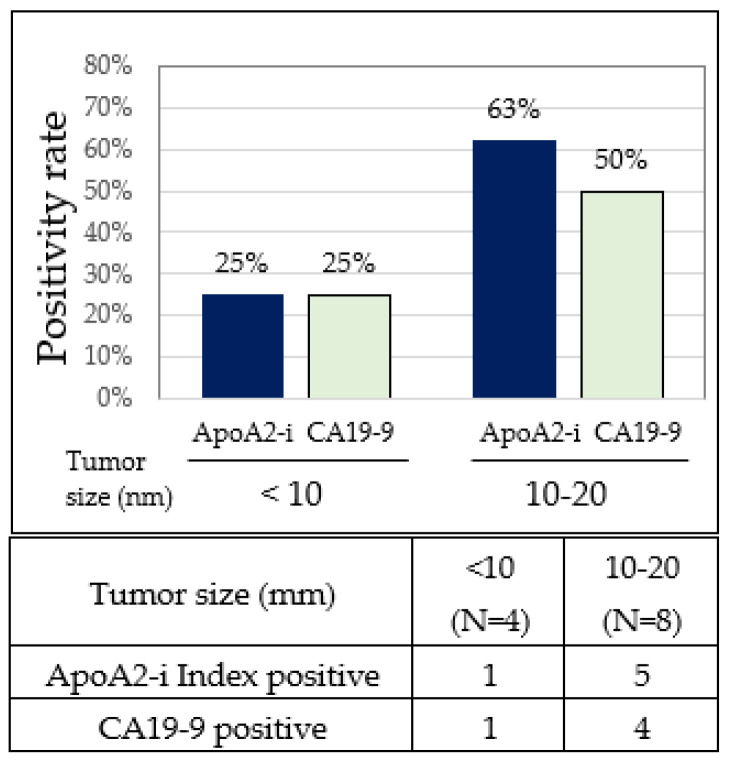
Positivity rates of apoA2-i Index and CA19-9 by stage IA PC. Positivity rates (**upper**) and number of positives (**lower**) for apoA2-i Index and CA19-9 in PCs with different tumor sizes.

**Figure 6 diagnostics-14-01920-f006:**
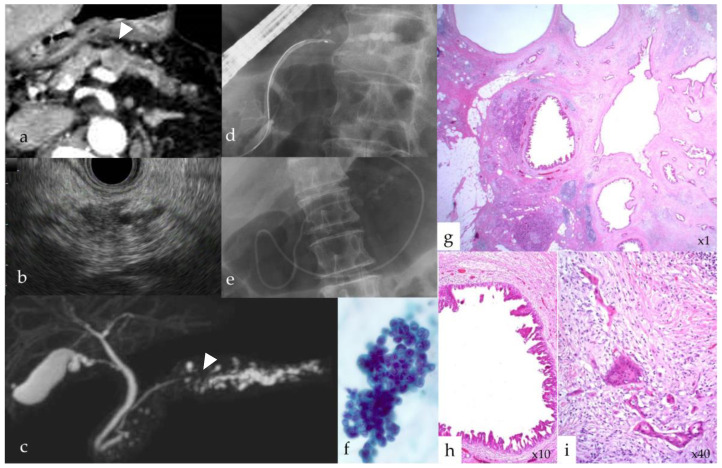
A case of a 78-year-old woman with stage IA pancreatic cancer (No. 13). (**a**) Computed tomography revealed a 15 mm tumor lesion (arrowhead) and local pancreatic atrophy in the pancreatic body and tail. (**b**) Endoscopic ultrasonography (EUS) revealed a hypoechoic area, indicating a mass with distal pancreatic duct dilatation. (**c**) Magnetic resonance cholangiopancreatography revealed stenosis of the main pancreatic duct (MPD; arrowhead) in the pancreatic body and dilatation in the caudal part. (**d**) Endoscopic retrograde cholangiography revealed stenosis of the MPD in the pancreatic body and dilatation in the caudal part. (**e**) An endoscopic nasopancreatic tube was inserted to collect pancreatic fluid for cytologic examination multiple times. (**f**) Pancreatic fluid cytology results were positive for adenocarcinoma. (**g**,**h**) Histopathological findings revealed high-grade pancreatic intraepithelial neoplasia (HG-PanIN) in the stenosis of the MPD, inflammation and fibrosis surrounding the HG-PanIN, and changes in fatty tissue in localized regions of the pancreatic parenchyma. (**i**) Invasive adenocarcinoma was observed around the HG-PanIN.

**Figure 7 diagnostics-14-01920-f007:**
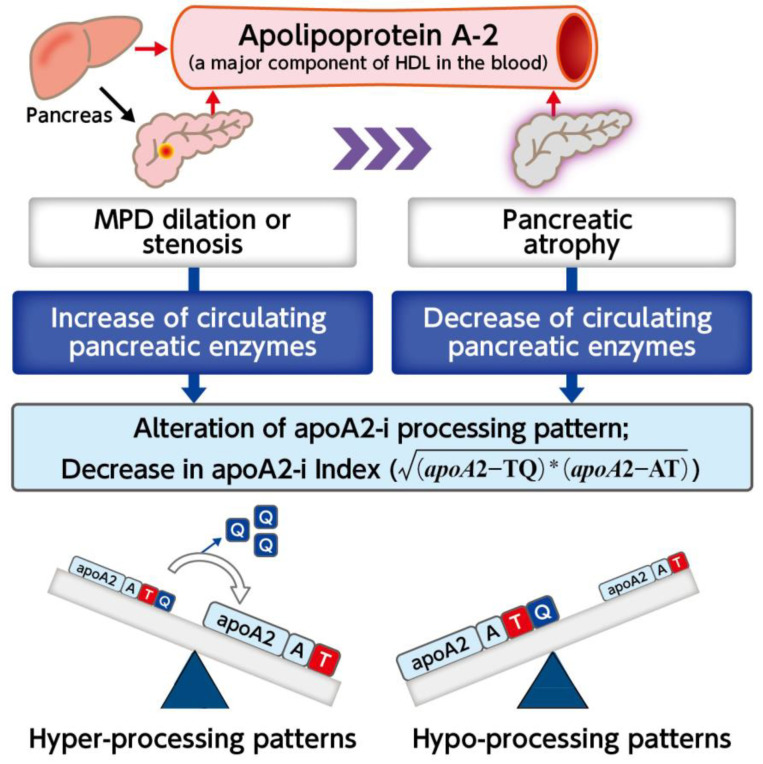
Possible mechanisms of apoA2-i production. In patients with pancreatic cancer, pancreatic exocrine function is impaired and levels of circulating pancreatic enzymes, such as pancreas-specific carboxypeptidase, are altered, leading to either hyper-processing or hypo-processing patterns. The hyper-processing pattern is caused by increased leakage of pancreatic enzymes into the bloodstream, whereas the hypo-processing pattern is caused by decreased production of pancreatic enzymes due to pancreatic devastation. Both patterns contribute to a decrease in the apoA2-i index. The asterisk stands for multiplication sign. HDL: high density lipoprotein.

**Table 1 diagnostics-14-01920-t001:** Patient lists and results in this study.

No.	Sex	Age	Stage	apoA2-i Index	CA19-9 (U/mL)	Intraductal Tumor Spread (mm)	Pathological Diagnosis	Imaging Findings			
								Tumor Lesion	MPD Dilatation by MRCP	MPD Stenosis by MRCP	Pancreatic Atrophy by CT
1	Female	58	0	92.1	5.8	5	HG-PanIN	No	Yes	No	Wide
2	Female	60	0	88.1	7.1	5	HG-PanIN	No	No	Yes	Local
3	Female	79	0	111.9	7.6	7	HG-PanIN	No	No	Yes	Local
4	Female	76	0	90.2	2 >	8	HG-PanIN	No	No	Yes	Local
5	Female	67	0	107.0	2 >	4	HG-PanIN	No	Yes	Yes	Local
6	Female	75	0	75.6	6.3	5	HG-PanIN	No	Yes	No	Wide
7	Female	58	0	91.5	20.3	3	HG-PanIN	No	Yes	Yes	Local
8	Female	66	0	73.9	2.6	3	HG-PanIN	No	Yes	Yes	Local
9	Male	69	0	151.8	12.0	10	HG-PanIN	No	Yes	No	Local
10	Male	71	0	52.0	44.2	10	IPMN with high grade dysplasia	Yes (Mural nodule) 10 mm	Yes	No	Wide
11	Male	86	0	55.3	4.9	5		Yes (Mural nodule) 7 mm	Yes	No	Local
12	Male	78	0	83.7	6.3	4, 3		Yes (Mural nodule) 5 mm	Yes	No	No
						Tumor size (mm)					
13	Female	78	IA	49.9	6.8	15	Well	Yes	Yes	Yes	Local
14	Female	78	IA	88.0	2 >	12, 12	Well	Yes	Yes	Yes	Local
15	Female	76	IA	58.2	38.6	15	Well	Yes	Yes	Yes	Wide
16	Female	83	IA	73.3	3.0	5	Well	Yes	Yes	Yes	Local
17	Female	79	IA	0.0	150.3	10	Well	Yes	Yes	Yes	Local
18	Male	82	IA	63.3	2.7	8	Well	Yes	Yes	Yes	Wide
19	Male	65	IA	42.9	9.9	4, 20	Well	Yes	Yes	Yes	Wide
20	Female	80	IA	100.8	2.5	16	Well	Yes	Yes	Yes	Local
21	Female	69	IA	75.4	162.0	20, 8	Mod	Yes	Yes	No	No
22	Male	83	IA	57.9	92.6	20	Mod	Yes	Yes	Yes	Wide
23	Female	75	IA	74.9	80.4	4	IPMN-derived invasive carcinoma	Yes	Yes	No	No
24	Female	79	IA	62.1	13.2	2	ITPN-derived invasive carcinoma	No	No	Yes	No

MPD, main pancreatic duct; MRCP, magnetic resonance cholangiopancreatography; CT, computed tomography; HG-PanIN, high-grade pancreatic intraepithelial neoplasia; Well, well-differentiated adenocarcinoma; Mod, moderately differentiated adenocarcinoma; IPMN, intraductal papillary mucinous neoplasm; ITPN, intraductal tubulopapillary neoplasm.

## Data Availability

The data presented in this study are available upon request from the corresponding author. The data are not publicly available due to privacy issues.

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
