# Peer review of "Potential of Carbohydrate Antigen 19-9 and Serum Apolipoprotein A2-Isoforms in the Diagnosis of Stage 0 and IA Pancreatic Cancer"

_diagnostics, 2024, doi:10.3390/diagnostics14171920_

Round 1

Reviewer 1 Report

Comments and Suggestions for Authors

I have read with great interest the original research article proposed by Keiji Hanada and colleagues. The argument treated by the Authors in the manuscript entitled “Potential of carbohydrate antigen 19-9 and serum apolipoprotein A2-isoforms in the diagnosis of stage 0 and IA pancreatic cancer” is of great interest for the scientific community and for pancreatic cancer research.

To my opinion there are minor aspects that should be elucidated.

The Authors should report in detail the protocol of plasma sample collection together with the protocol of ELISA test to help the reproducibility of the experiments.

Are the Elisa kit and the specific Antibodies commercially available?

In the diagnosis of early pancreatic cancer, have the Authors routinely indicate the EUS-guided fine needle aspiration and fluid cytology?

Please, might the Authors discuss the following aspects:

What is the percentage of patients with chronic pancreatitis associated with pancreatic cancer?

Might the chronic inflammation alter the serum levels of apoA2 isoforms?

Are the apoA2 isoforms specific only in early stage of carcinogenesis or not?

Might the Authors graphically design a prognostic curve according to the ApoA2 serum levels?

There are minor grammatic revisions.

Thank you for your proposed manuscript.

Author Response

The reply is attached.

Reviewer 2 Report

Comments and Suggestions for Authors

This study simply determined the concentration of serum apolipoprotein A2 isoforms through ELISA to establish it as a new biomarker for the diagnosis of pancreatic cancer (PC).

In my opinion, other approaches and techniques should have been applied to support the results. (i) The authors should have purified apolipoprotein A2  isoforms, performed its sequencing, and also conducted Western blotting. Additionally, the study should have been performed on mice, inducing PC and then examining whether apolipoprotein A2 isoforms also displayed same positivity rates.

The authors should also unveil the underlying molecular mechanism explaining how apolipoprotein A2 isoforms are involved in PC.

In Line 12 and Line 17, please write out the full forms of the abbreviations first.

I suggest the authors draw flow diagram representing each step of the study for better understanding of the readers.

Finally present the data in the form of figures as well.

Author Response

The reply is attached.

Round 2

Reviewer 2 Report

Comments and Suggestions for Authors

Manuscript is recommended for publication.